# Applicability Assessment of Passive Microwave LST Downscaling over Semi–Homogeneous Desert Underlying Surface Based on Machine Learning

Yongkang Li [1,2,3], Yongqiang Liu [1], Wenjiang Huang [4], Yang Yan [5], Jiao Tan [1] and Qing He [2,3,*]

1   College of Geography and Remote Sensing Sciences, Xinjiang University, Urumqi 830052, China; yongkangl@stu.xju.edu.cn (Y.L.); liuyq@xju.edu.cn (Y.L.); tangis@stu.xju.edu.cn (J.T.)
2   Institute of Desert Meteorology, China Meteorological Administration, Urumqi 830002, China
3   Field Scientific Experiment Base of Akedala Atmospheric Background Station, China Meteorological Administration, Altay 836500, China
4   State Key Laboratory of Remote Sensing Science, Aerospace Information Research Institute, Chinese Academy of Sciences, Beijing 100094, China; huangwj@aircas.ac.cn
5   College of Resaurces and Environment, Xinjiang Agricultural Unversity, Urumqi 830052, China; 320202394@xjau.edu.cn
*   Correspondence: qinghe@idm.cn

**Abstract:** The spatial and temporal resolution of remote sensing products in land surface temperature (LST) studies can be improved using the downscaling method. This is a crucial area of research as it provides basic data for the study of climate change. However, there have been few studies evaluating the applicability of downscaling methods using underlying surfaces of varying complexities. In this study, we focused on the semi–homogeneous underlying surface of Gurbantunggut Desert and evaluated the applicability of five classical, passive microwave, downscaling methods based on the machine learning of Catboost, using 365 days of AMSR–2 and MODIS data in 2019, which can be scanned once during the day and night. Our results showed four main points: (1) The correlation coefficients between feature vectors and the LST of the semi–homogeneous underlying surface were clearly different from those of the surrounding oases. The correlation coefficient of the semi–homogeneous underlying surface was high, and that of the surrounding oases was low. (2) At the same frequency, the correlation coefficient between vertically polarized BT and LST was greater than that between horizontally polarized BT and LST. Considering the semi–heterogeneous underlying surface, 23.8 GHz and 36.5 GHz may be more suitable for passive microwave LST retrieval than 89 GHz according to physical mechanisms. (3) The fine–scale LST downscaling accuracy achieved with all BT channels of AMSR–2 was higher than that achieved with the other four classical models. The day and night RMSE values verified with MYD11A1 data were 2.82 K and 1.38 K, respectively. (4) The correlation coefficients between downscaled LST and the soil temperature of the top layer of the site were the highest, with daytime–nighttime $R^2$ values of 0.978 and 0.970, and RMSE values of 3.42 and 4.99 K, respectively. The all–channel–based LST downscaling method is very effective and can provide a theoretical foundation for the acquisition of all–weather, multi–layer soil temperature.

**Keywords:** land surface temperature; AMSR–2; Catboost; downscale; Gurbantunggut Desert





## 1. Introduction

Land surface temperature (LST) has been listed as one of the priority parameters in the International Geospheric Biosphere Programme (IGBP) [1] and was recognized as one of the 54 essential climate variables (ECVs) by the Global Climate Observing System (GCOS) [2,3]. It has been widely used in the fields of urban heat island effect [4,5], fire monitoring [6], climate monitoring [7,8], energy balance assessment [9,10], and evapotranspiration [11–13],

among other research fields [14]. Due to the complexity of topography, soil, vegetation, weather, and other factors, the LST is heterogeneous spatially [15,16] and fluctuates in time; therefore, to accurately describe these variations, high spatial and temporal resolution data are necessary. Currently, thermal infrared and optical remote sensing clear sky LST retrieval algorithms, such as MODIS and AVHRR, have achieved high accuracy [17–19]. However, in non–clear sky areas, such as those affected by clouds, the accuracy of these retrieval algorithms is seriously compromised [20,21]. To overcome the limitations of optical remote sensing, microwave remote sensing is used to obtain surface radiation information under complex weather conditions [22,23]. The spatial resolution of passive microwave remote sensing is low, but it can penetrate clouds and is weakly affected by the atmosphere [24]. Therefore, passive microwave remote sensing is ideal for all–weather observations of LST. However, to improve the spatial resolution for regional scale research, other data sources must be fused with the passive microwave data. The passive microwave LST spatial down-scaling technique enables the acquisition of more detailed LST information, even under complex weather conditions [25,26]. The passive microwave–based downscaling method is a data fusion technique that relies solely on remote sensing and spectral information to enhance the spatial resolution of passive microwave remote sensing. It utilizes high spatial resolution auxiliary data to reveal the spatial heterogeneity of the land surface, and it provides more accurate descriptions of the spatial distribution of LST. Unlike distance–based spatial interpolation methods, the downscaling method does not estimate unknown pixels based on the distance to known sample points. Instead, it determines pixel values from low spatial resolution passive microwave data, and then fuses them with high spatial resolution LST information to generate more accurate high–resolution LST data. It can provide more accurate and reliable LST, which is crucial for climate change research, natural resource management, and environmental monitoring.

　　Improving the spatial and temporal resolution of remote sensing products using the downscaling method has always been a hot topic in LST study, as it can provide basic data for the study of climate change. However, the suitability assessment of downscaling methods with a particular type of underlying surface is often lacking. The considered study area is always a complex mixed surface. Thus, it is urgent to investigate downscaling methods using transitional heterogeneous surfaces, which can provide a basis for the study of complex heterogeneous surfaces. For instance, Benjamin et al. [27] downscaled the LST of an urban area of Hamburg, Germany, to 100 m spatial resolution. Christopher and Michael [28] obtained the 250 m spatially resolved LST, downscaled from 1000 m resolution, of heterogeneous regions of the Eastern Mediterranean that contained forests, mountains, and deserts. Li et al. [29] downscaled the MODIS LST product of rural and urban areas of Beijing that contained vegetation, croplands, buildings, and roads from 990 to 90 m resolution. Samuel et al. [30] calibrated the 2500 m SSMIS passive microwave brightness temperature data to 5000 m land surface temperature using a statistical down-scaling method for a region in Southern Africa. Microwave channels can penetrate clouds, are weakly affected by the atmosphere, and can obtain surface radiation information in complex weather [22,23,31,32]; however, the spatial resolution of passive microwaves is low. The fusion and downscaling of passive microwaves to obtain continuous LST with high spatio–temporal resolution is a promising field of research.

　　Statistical models have been developed to solve the LST ill–conditioned retrieval problem of multi–channel brightness temperature (BT) [32,33], which is caused by the parameter of surface roughness, and the attenuation and extinction of vegetation [34–36]. McFarland et al. [37] eliminated the water and snow cover in the study area; then, they corrected the influence of water vapor using other channels and retrieved the LST using multiple linear regression by taking an SSM/I (special sensor microwave/imager) as the data source and 37 GHz as the most important regression factor. Then, Gao [36], Holmes [38], and Njoku et al. [39] used AMSR–E (Advanced Microwave Scanning Radiometer for EOS) data to analyze the LST inversion. Their studies showed that the vertically polarized BT channel at 36.5 GHz is the optimal band for LST inversion, because it is slightly influenced by

the atmosphere. However, they also found that the low soil moisture and the scattering effect of bare soil surface can reduce the accuracy of LST inversion due to passive microwaves. Mao et al. [32] used AMSR–E data and conducted linear regression analyses on BT at all AMSR–E frequencies and MODIS LST data of different surface types. They found that the 89 GHz vertically polarized BT had the highest correlation and that low correlation existed between low–frequency BT and LST. Retrieval frameworks combining the advantages of thermal infrared (TIR) and passive microwave remote sensing have been proposed and are continuously being refined to generate LST at all–weather, real–time resolution [40,41]. Traditional studies of passive microwave BT and LST are typically simplified using characteristic bands or physical models; they do not fully use the information provided by multiple bands. This limitation can be overcome using machine learning methods, which can provide simpler solutions to non–linear multi–variate problems and significantly improve inversion accuracy [24,33,42].

Machine learning is an effective method for mining the relationship between multi–channel brightness temperature and LST. However, there are many challenges in the inversion of the passive microwave surface temperature of complex and heterogeneous surfaces, such as mountains. A few attempts at passive microwave LST downscaling using semi–heterogeneous underlying surfaces using machine learning, which can provide relevant methodological and theoretical support for complex and heterogeneous surfaces, have already been conducted. Gurbantunggut Desert is a fixed and semi–fixed desert and was here selected as a typical semi–heterogeneous underlying surface. In terms of complexity, it is a transitional form between homogeneous surfaces (e.g., shifting sand desert) and other complex underlying surfaces (e.g., mountains). Categorical boosting (Catboost) is a new gradient boosting decision tree (GBDT) machine learning algorithm proposed by Yandex in 2017, which can improve the overfitting problem of previous GBDT algorithms (such as LightGBM and XGboost) [43]. Catboost has higher accuracy than random forests and support vector machines, and consumes less computational resources [43]. Its robustness has gradually begun to be applied in studies such as evapotranspiration inversion [44,45], weather forecasting [46,47], and disease analysis [48,49].Therefore, Gurbantunggut Desert is a natural testing ground for the applicability evaluation of the downscaling of the microwave LST of a semi–homogeneous ground surface using the machine learning of Catboost.

To fill these gaps in the research, we attempted to answer the following question: using passive microwaves, how can we develop a downscaling model of a semi–homogeneous underlying surface and obtain all–weather, fine–scale LST? Specifically, this study aimed at (1) determining the relationship between the BT information of AMSR–2 multi–channel passive microwaves and LST; (2) constructing an LST downscaling model based on Catboost using AMSR–2 and validating its stability in developing fine–scale LST; (3) obtaining an optimal LST downscaling model by comparing the retrieval accuracy of the cross–validation results using MYD11A1 products; and (4) comparing the result correlation and error with the site multi–layer soil temperature and mapping the all–weather, diurnal–scale, fine–scale LST of the semi–heterogeneous underlying surface. In Section 2, we introduce the characteristics of the study area, the data sources used (including site data and remote sensing data), the data processing procedures, and the methods used. In Section 3, we provide a comprehensive introduction to the results of the paper and explain the relevant phenomena. Specifically, Section 3.1 shows the correlation between the characteristic factors of semi–heterogeneous underlying surfaces and land surface temperature, which provides a preliminary evaluation of the selection of each factor and its contribution to the results. Section 3.2 shows the 10–fold cross–validation results of five classic models during the training process. Section 3.3 conducts cross–validation of the five classic downscaled models using MYD11A1's LST error $\leq 1$ K per pixel, selects the optimal model, and analyzes the contribution of each characteristic factor to the Catboost model. Section 3.4 analyzes the relationship between the downscaled results of passive microwave land surface temperature and multi–layer soil temperature using data from the unique measured station in

the study area. Section 4 quantitatively discusses the accuracy of the current downscaled results and related research, and it provides prospects for future work. Section 5 presents the main conclusions of the paper.

## 2. Method

### 2.1. Study Area

Gurbantunggut Desert is the second largest desert in China, covering an area of about 48,800 km$^2$ [50]. It is located in the middle of Junggar Basin and is surrounded by the Tianshan Mountains in the south, Junggar Mountains in the west, Beitashan Mountains in the east, and Altai Mountains in the north (Figure 1). It is a typical fixed and semi–fixed desert, where the most representative sand dune type is sand ridge, accounting for 80% of the desert area [51]. It presents a typical continental arid climate in the mid–temperate zone that is controlled by the westerly belt all year round; as a result, the longitudinal sand dunes move from northwest to southeast. In Gurbantunggut Desert, the annual evaporation is 2000~2800 mm; the annual precipitation does not exceed 220 mm; and the desert hinterland is only 70~100 mm, concentrated in May–September [52–54]. Stable snowfall in winter and spring provides favorable moisture conditions for vegetation growth in desert areas. There are short–lived plants vigorously growing in spring and summer. In Gurbantunggut Desert, the day–night cloud coverage in 2019 reached 63.0% and 65.9%, respectively, according to MYD11A1 data. The temporal and spatial continuity was significantly affected.

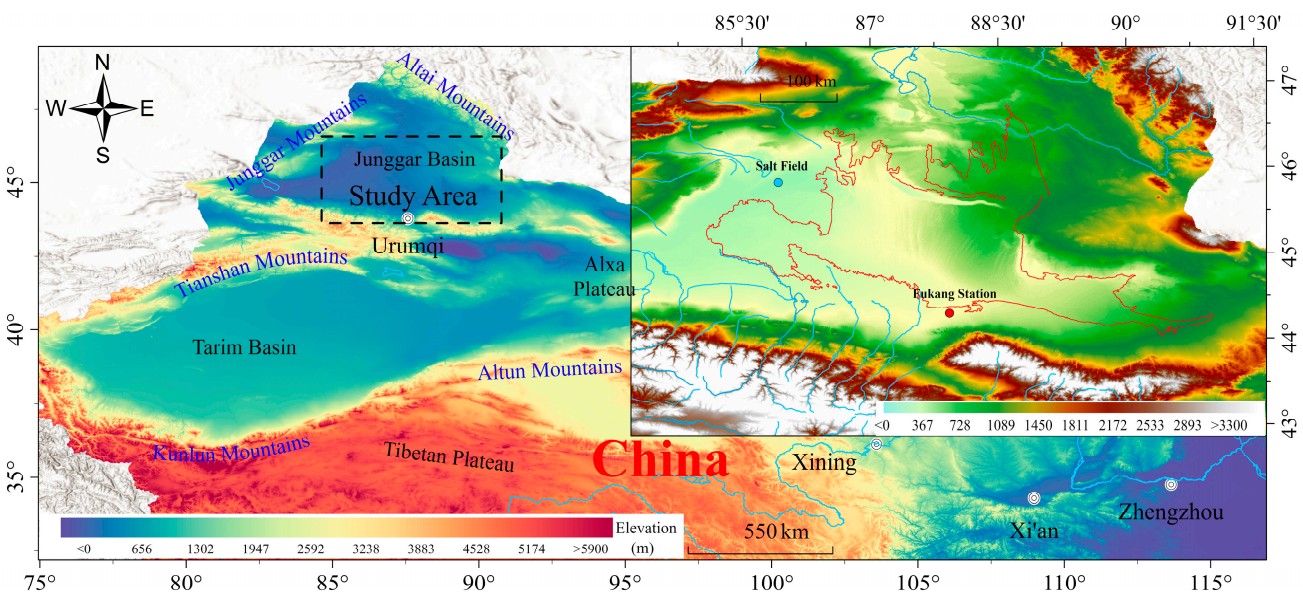

**Figure 1.** Overview of study area.

The heterogeneity and complexity of underlying surfaces can affect the accuracy of LST retrieval and the applicability of downscaling methods. Shifting sandy lands without vegetation are relatively homogeneous and have open underlying surfaces, e.g., Taklimakan Desert, which makes them a suitable verification field for LST retrieval. There is little practical value in conducting downscaling work in this region. Due to the heterogeneity and complexity of mountainous underlying surfaces, the application and evaluation of passive microwave LST downscaling face great challenges. Since Gurbantunggut Desert presents periodic growth of vegetation and fluctuation of sand dunes, it can be considered a transition form from homogeneous to complex underlying surfaces. Downscaling studies in this region are of great importance for mountainous areas.

The cost of conducting experiments in Gurbantunggut Desert is high, causing a chronic lack of empirical data on key surface parameters, such as LST. This poses great difficulties in the validation of surface parameters in Gurbantunggut Desert. Daily, hour–by–hour

LSTs are currently only available at Fukang Desert Ecological Station at the southern edge of the oases (http://rs.cern.ac.cn/index.jsp (accessed on 7 December 2020)).

*2.2. Datasets*

2.2.1. Remote Sensing Data

The 2019 AMSR–2 sensor L1R and MYD13A3 datasets, which were obtained by the GCOM–W and Aqua satellites, respectively, were used as feature vectors for Catboost model inputs. MYD11A1 data were used as target vectors for model training.

JAXA (http://gportal.jaxa.jp (accessed on 9 May 2020)) provides passive microwave bright temperature AMSR–2 data from June 2012 to February 2023, comprising a total of 14 channels vertically and horizontally polarized at 6.9 GHz, 10.7 GHz, 18.7 GHz, 23.8 GHz, 36.5 GHz, and 89.0 GHz. The daily equatorial transit times of AMSR–2 are approximately 1:30 (descending orbit) and 13:30 (ascending orbit). AMSR–2 conducts two scans per day, covering approximately 90% of the globe. The ascending and descending orbit data cover most of the globe, except for the poles, within 2 d. In a single scan cycle (16 days), there are 14 days that the satellite transits above Gurbantunggut Desert at approximately 13:00 during the day and 1:00 at night.

LADDC provides MYD13A2 and MYD11A1 data from July 2002 to February 2023. MYD13A2 are 16–day synthetic enhanced vegetation index (EVI) and normalized vegetation index (NDVI) products. MYD11A1 are daily LST data. AMSR–2 passive microwave BT data and MYD–related products are acquired at similar times, with daily transit time differences of <15 min within a single scan period (16 days) in Gurbantunggut Desert [25]. MYD11A1 data have a significant lack of spatial continuity due to cloud influence. The MYD11A1 LST data in the Gurbantunggut Desert region are significantly impacted by cloud cover, with cloud cover percentages of 63.0% during the day and 65.9% at night for the MODIS LST products in 2019. This has resulted in marked temporal and spatial discontinuities, greatly hindering data utilization and further research. To address this issue and improve the temporal and spatial continuity of the LST product, it is necessary to integrate microwave data. Therefore, MYD11A1 pixels with quality control average LST error $\leq 1$ K, which are not affected by clouds, were selected to establish a mapping relationship with AMSR–2 passive microwave bright temperature. Detailed data descriptions are given in Table 1.

**Table 1.** Information of AMSR–2 microwave brightness temperature and MODIS data.

| AMSR–2 Passive Microwave Bright Temperature Data | | | MODIS Data | | |
|---|---|---|---|---|---|
| Center Frequency (GHz) | Polarization Direction | Spatial Resolution (km) | Data Types | Spatial Resolution (km) | Dataset |
| 6.925/7.3 | V/H | 10 | | | |
| 10.65 | V/H | 10 | MYD 11A1 | 1 | LST_Day_1 kmQC_Day |
| 18.7 | V/H | 10 | | | |
| 23.8 | V/H | 10 | | | |
| 36.5 | V/H | 10 | MYD 13A2 | 1 | 1 km_16_days_EVI1 km_16_days_NDVI |
| 89 | V/H | 5 | | | |

Note: H, horizontal polarization; V, vertical polarization.

2.2.2. Ground–Measured Data

Fukang Desert Ecological Station is located at 87°33′36″E and 44°10′30″N, at the southern edge of Gurbantunggut Desert. It is a transition zone between oases and desert, with lush ground vegetation, mostly desert grassland and meadow grassland [55–57]. The station is at an elevation of 461 m and provides daily, hour–by–hour, 6–layer soil temperature data (5, 10, 20, 40, 80, and 160 cm below the surface). Due to the presence of the AMSR–2 scan gap and the absence of land temperature collection at the automatic station, a total of 253 data were selected for LST validation. Automatic station data were provided

by National Center for Ecological Sciences (http://rs.cern.ac.cn/index.jsp (accessed on 7 December 2020)).

2.2.3. Data Processing

According to previous studies [24,25,32,42], five Catboost downscaling models were constructed based on the combination of passive microwave BT and MYD13A2 vegetation index data. Then, 10 km resolved passive microwave bright temperature was downscaled to 1 km resolved LST. In our study, the Catboost machine learning algorithm was used instead of the traditional statistical model to construct the spatial structural similarity of 10 km and 1 km passive microwaves. The steps for the downscaling of the AMSR–2 daytime and nighttime LST of the considered semi–heterogeneous underlying surface were as follows (Figure 2):

(1) AMSR–2 and MODIS data were resampled to 10 km × 10 km (D1). Then, MYD11A1 pixels with quality average LST error ≤ 1 K were selected to match AMSR–2 microwave BT data and MYD13A2 vegetation index data.

(2) The spatial correlation of the feature vectors was statistically analyzed. In order to obtain the spatial pattern of correlation coefficients (r) and probability (p) for each feature vector, 14 BTs and 2 vegetation indices were correlated with MYD11A1 data at a resolution of 10 km. The frequency distribution statistics of the correlation coefficients were calculated. On this basis, the frequency distribution characteristics and spatial characteristics of the correlation between each feature vector and LST data were analyzed.

(3) Downscaled combinations of feature vectors were constructed. Based on the statistical analysis of correlation and the research studies of previous scholars, feature vector combinations were selected to train the Catboost model. The feature combinations were 7–channel algorithm [24] (difference between vertical polarization and horizontal polarization (V–H)), horizontal polarization combination [42] (7 horizontally polarized channels (H)), vertical polarization combination [42] (7 vertically polarized channels (V)), semi–empirical combination [32] (36.5 V/36.5 V–23.8 V/36.5 V–18.7 H/89 V GHz (Phy)), and full channel combination [42] (14 channels (VH)), respectively. The five microwave vector combinations were combined with vegetation indices to train five models of machine learning. The five models were initially evaluated using "10–fold cross–validation" to avoid overfitting [58–60].

(4) Passive microwave BT data were resampled to 1 km (D2) using the nearest neighbor algorithm. Then, the 1 km MYD13A3 vegetation index data of the corresponding pixels were composed of five sets of feature vectors, which were input to the five machine learning models at 10 km resolution, respectively. The output 1 km pixels were the passive microwave surface temperature downscaling results.

(5) MYD11A1 quality average LST error ≤1 K clear–sky pixels were selected to assess the consistency of downscaled surface temperature and MODIS data. Ten–fold cross–validation and MODIS validation were used to determine which was the optimal combination of downscaling feature vectors for the LST of a semi–heterogeneous underlying surface.

(6) Six–layer soil temperature data from Fukang station were selected for accuracy verification both to determine the correlation between soil temperature and passive microwave LST downscaling for each layer and to evaluate the optimal model. The importance of each feature vector in the optimal model was also summarized and analyzed in this last step.

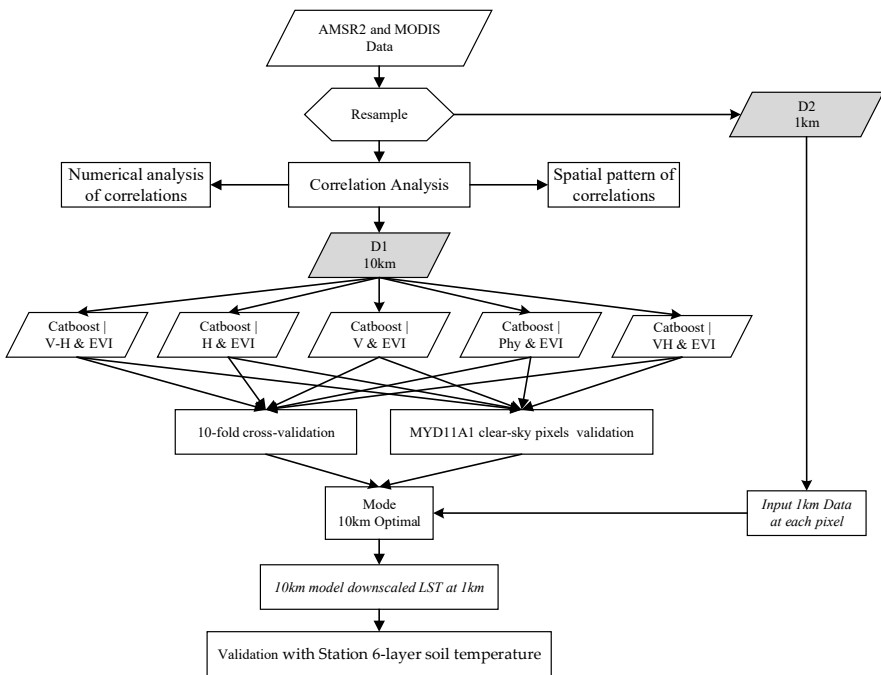

**Figure 2.** Flowchart of data processing steps.

*2.3. Methods*

2.3.1. Microwave Radiation Transmission Theory—Radiative Transmission Model for Microwave Surface Temperature Inversion

The physical basis for the passive microwave observation inversion of LST is radiative transfer theory [24,32]. The LST retrieval algorithm is established according to radiative energy balance. The radiative transfer equation describes the total energy received by the passive microwave radiometer, including surface radiation, and upward and downward atmospheric path radiation, as well as the radiation component attenuated by atmospheric absorption [32,33]. The general radiative transfer equation is expressed as follows:

$$B_f\left(T_f\right) = \tau_f(\theta)\varepsilon_f B_f(T_s) + \left[1 - \tau_f(\theta)\right](1-)\tau_f(\theta)B_f\left(T_a^{\downarrow}\right) + [1 - \tau_i(\theta)]B_f\left(T_a^{\uparrow}\right) \quad (1)$$

where $T_s$ represents the surface temperature, $T_a$ represents the mean atmospheric temperature, $T_f$ represents the brightness temperature at the $f$ frequency, $\tau_f(\theta)$ represents the transmittance at the $f$ frequency at the observation angle, $\varepsilon_f$ represents the surface emissivity at the $f$ frequency, $B_f(T_s)$ represents the surface emission, and $T_a^{\downarrow}$ and $T_a^{\uparrow}$ are atmospheric downlink and uplink radiation, respectively.

The inversion of LST based on the multi–channel radiative transfer equation adds a new unknown for each added channel, i.e., there are N + 1 unknowns when N channels are added; therefore, the inversion of LST based on the physical model of passive microwaves is an ill–conditioned retrieval problem [34,42]. To solve this problem, Fily et al. [35] assumed that the emissivity of vertical polarization and horizontal polarization at the same frequency can be simplified to a linear relationship, $\varepsilon_V = a\varepsilon_H + b$. Thus, Equation (1) can also be expressed as follows:

$$LST = \left[T_{b_V} - aT_{b_H} - (1 - b - a)\tau_f(\theta)T_a^{\downarrow}\right] / \left[b\tau_f(\theta)\right] \quad (2)$$

where $\varepsilon_{V/H}$ represents the surface emissivity in the vertically and horizontally polarized directions and $a, b$ are linear regression coefficients. However, this assumption simplifies preconditions such as constant atmospheric correction, absorption by surface vegetation, and the overlooking of atmospheric scattering effects [34–36].

### 2.3.2. Categorical Boosting—Catboost

The machine learning method for the multi–channel microwave BT inversion of LST can avoid the simple linearization of the vertical and horizontal BT inter–relationship. Thus, it can better solve the ill–conditioned retrieval problem. Catboost is a newly proposed gradient boosting decision tree (GBDT) algorithm based on Xgboost, LightBMG, etc. [43], that can better solve complex data heterogeneity, high noise, feature interdependence, and other problems [61]. In 2019, Huang et al. [44] used Catboost to study daily scale evapotranspiration in wet areas in China, which was the first application of Catboost to remote sensing field research. The study showed that Catboost is better than random forest and support vector machine in terms of accuracy and stability. In the Catboost algorithm, feature vectors provide more information than target vectors [43], while the traditional GBDT algorithm uses the average target vector as the node segmentation criterion [45]. Consequently, Catboost can exploit the connection between feature vectors, enrich the feature dimension, and reduce the impact of frequency target vectors on the results. Catboost is a greedy machine learning approach that combines all features and uses all feature combinations to build decision trees [61]. In addition, it overcomes the overfitting problem of traditional GBDT algorithms by sorting gradient boosting, adding prior distribution terms, and removing noisy data from the training set. It can solve the prediction gradient bias [43].

In the practical application of Catboost, greedy learning leads to statistic and grouping target variables at each iteration. Its dense numerical features lead to the high computational burden of building decision trees. The above problems of the Catboost algorithm can be overcome with a GPU in the python interface, which supports multiple working GPU threads. Distributed tree learning can parallelly process data or features. AMSR–2 passive microwave BT data have multi–frequency, dual–polarization characteristics. The Catboost algorithm can ensure the extraction of multi–band features while improving the learning efficiency.

### 2.3.3. Validation Methods

In this study, the mapping relationship between AMSR–2 passive microwave BT and MODIS LST was constructed by training five Catboost models. The "10–fold cross–validation" method was selected to initially evaluate the machine learning mode [58–60]. Using "10–fold cross–validation", in our study, training data on Gurbantunggut Desert were randomly divided into ten groups. Nine of them were cyclically used as training data to fit the model. The remaining group was used for validation. The mean metric of the fitting accuracy obtained ten times was taken as the final accuracy of this model. The use of 10–fold cross–validation can effectively reduce systematic error, random error, and coarse error to achieve the theoretical maximum precision of a machine learning model [62]. It helps to reduce the risk of overfitting and provides a more accurate estimate of the model's performance.

The Pearson correlation coefficient was used to analyze the correlation between each feature vector and MYD11A1 data. Consequently, the correlation coefficient (R) and probability (P) were obtained for each pixel, and we analyzed the spatial patterns and frequency distribution of the correlation coefficients. The physical significance of different frequency polarization modes in LST was analyzed with the correlation coefficient and feature importance using the considered semi–heterogeneous underlying surface. The coefficient of determination ($R^2$), mean absolute error (MAE), and root mean square error (RMSE) were used as the evaluation metrics of the Catboost model. Hyperparameter search, verification, and comparison were carried out based on these metrics.

## 3. Result

### 3.1. Correlation Statistical Analysis of Feature Vectors and LST

We analyzed the correlations between 16 feature vectors, and daytime and nighttime LST. These 16 feature vectors were needed to construct the five Catboost models.

Our analysis showed that the correlations between feature vectors and LST exhibited significant differences between daytime and nighttime (Figure 3). The 25th and 75th percentiles of correlation are represented by the red solid lines, while the dashed line represents the median of the correlation statistics. Notably, the correlation between horizontally polarized BT and LST was higher during the daytime than at night, especially for passive microwave BT. Furthermore, this relatively high trend decreased with an increase in frequency. This difference between daytime and nighttime data, along with increasing frequency correlation characteristics, also appeared in the correlation between vertically polarized BT and LST.

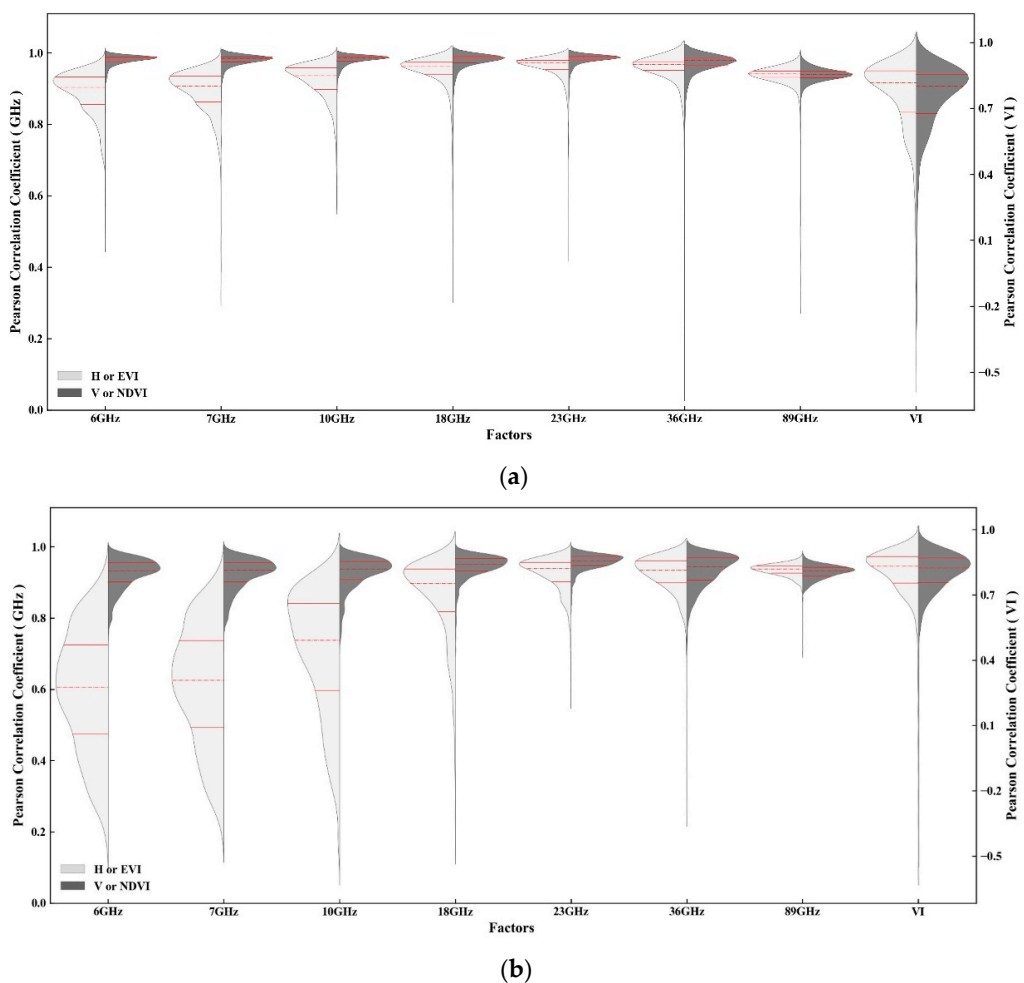

**Figure 3.** Histogram of correlations between 16 feature vectors and LST. (**a**) Daytime. (**b**) Nighttime.

In addition, each BT of passive microwaves and the vegetation index data were positively correlated with LST. The correlation of vertical polarization was greater than that of horizontal polarization at the same frequency, and the EVI correlation with LST was also greater than that of NDVI. Except for the correlation at 89 GHz, the value distribution characteristics of the correlation for nighttime data were more discrete than those for daytime data, and the characteristics of horizontal polarization were more discrete than those of vertical polarization.

In terms of spatial patterns of correlation, the semi–heterogeneous underlying surface of Gurbantunggut Desert was greatly different from the surrounding oases (Figure 4). The difference in the correlation daytime spatial patterns was clearer than that in the correlation nighttime special patterns. The correlation level of the semi–heterogeneous underlying surface in the interior of the desert was higher than that of the oases surrounding the desert.

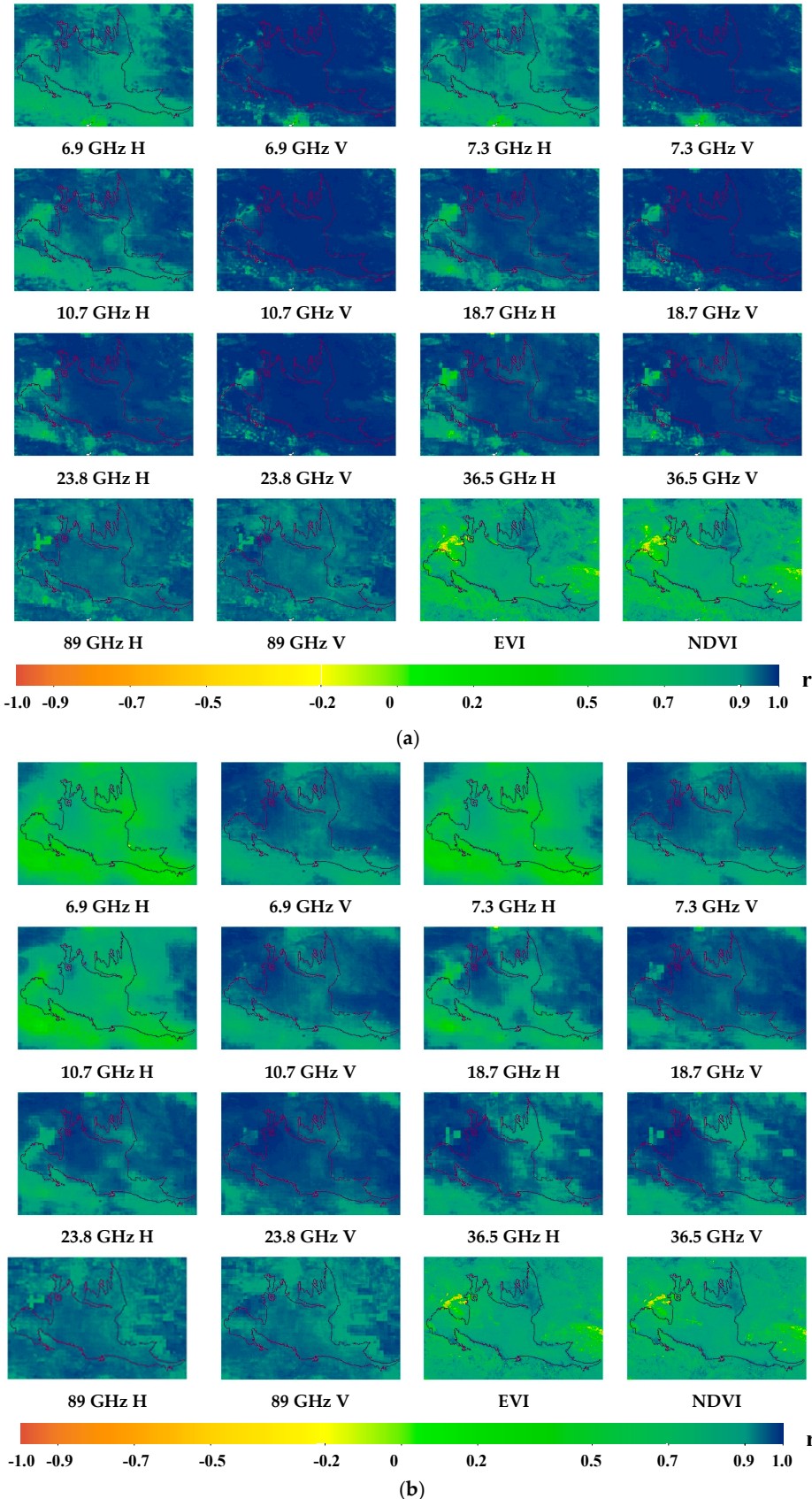

**Figure 4.** Spatial patterns of correlations between 16 feature vectors and LST. (**a**) Daytime. (**b**) Nighttime.

From the histogram of 16 feature vectors, we found that the correlation coefficients between the BT at 89 GHz and LST were higher and their frequency distribution was more aggregated. However, the correlation distributions at 23.8 GHz and 36.5 GHz were more discrete. The low correlation values lowered the mean and median at 23.8 GHz and 36.5 GHz. The spatial pattern of correlation at 89 GHz was more homogeneous than those at 23.8 GHz and 36.5 GHz. Regarding the spatial patterns of correlation at 23.8 GHz and 36.5 GHz, the boundaries between oases and desert were clearer. In fact, BT at 23.8 GHz and 36.5 GHz were more correlated with LST than BT at 89 GHz in the interior of Gurbantunggut Desert. Therefore, 23.8 GHz and 36.5 GHz may be more suitable for LST downscaling with passive microwaves considering a semi–heterogeneous subsurface than 89 GHz from a physical point of view.

Next, we ranked the mean values of the correlation coefficients of each feature vector in the rectangular region (the bounding rectangle of the Gurbantunggut Desert area). Daytime—vertical: 23.6 GHz (0.980) > 6 GHz (0.977) = 10 GHz (0.977) > 7 GHz (0.975) > 18 GHz (0.972) > 36.5 GHz (0.962) > 89 GHz (0.936). Daytime—horizontal: 23.6 GHz (0.961) > 36.5 GHz (0.952) > 18 GH (0.944) > 89 GHz (0.932) > 10 GHz (0.922) > 7 GHz (0.891) > 6 GHz (0.883), EVI (0.761) > NDVI (0.742). Night—vertical: 23.6 GHz (0.957) > 18 GHz (0.943) > 36.5 GHz (0.932) > 89 GHz (0.929) > 10 GHz (0.926) > 6 GHz (0.923) = 7 GHz (0.923). Night—horizontal: 89 GHz (0.934) > 23.6 GHz (0.921) > 36.5 GHz (0.919) > 18 GHz (0.856) > 10 GHz (0.703) > 7 GHz (0.611) > 6 GHz (0.595), EVI (0.802) > NDVI (0.798). For vertical polarization, 23.6 GHz had the highest correlation with both daytime and nighttime surface temperatures; the lowest correlations were at 89 GHz (daytime), and 6 GHz and 7 GHz (nighttime). For horizontal polarization, the highest correlations were at 23.6 GHz and 89 GHz (daytime and nighttime, respectively); the lowest correlation was at 6 GHz (both daytime and at nighttime). The correlation statistics are consistent with the results of horizontal polarization presented by Kebiao Mao [33], where lower–frequency microwave BT had a lower correlation with LST.

Correlation statistical analysis was simultaneously performed at the pixel level with correlation coefficient r and probability p. Probability p was less than 0.01 at a highly significant level, except for the exposed salt field aggregation area in the northwest of the outer Gurbantunggut Desert. The results showed that the exposed salt mine significantly affected the mapping relationship between the BT of passive microwaves and LST.

In our study, the correlation of polarization difference at the same frequency was also investigated. The results showed that the correlation between polarization difference and LST at the same frequency tended to decrease as the frequency increased. The spatial distribution of the correlation between polarization difference and LST at 18.7 GHz could better distinguish the desert from the surrounding oases, with its boundary being clearer.

### 3.2. Ten–Fold Cross–Validation of Catboost for Five Feature Vector Combinations

The initial evolution of the generalization capability of the five Catboost models was performed using "10–fold cross–validation" on daytime and nighttime data. The difference among the five Catboost models (V–H and EVI, H and EVI, Phy and EVI, V and EVI, and VH and EVI) was the combination of the selected feature vectors. The optimal parameters of each feature vector combination were determined using a hyperparameter search. The training results of Catboost are shown in Figure 5.

The five combinations of feature vectors were used to construct daytime and nighttime LST inversion models with EVI data. The 10–fold cross–validation results show that the model performance followed the order, from high to low, of VH > V > H > Phy > V–H. The Catboost model based on VH showed optimal results, where daytime $R^2$, MAE, and RMSE were 0.992, 1.50 K, and 2.08 K, respectively. Nighttime $R^2$, MAE, and RMSE were 0.993, 0.82 K, and 1.30 K, respectively. This indicted that Catboost based on VH could establish the mapping relationship between the BT of passive microwaves and LST better than the other feature vector combinations and showed robustness for the semi–heterogeneous underlying surface of Gurbantunggut Desert.

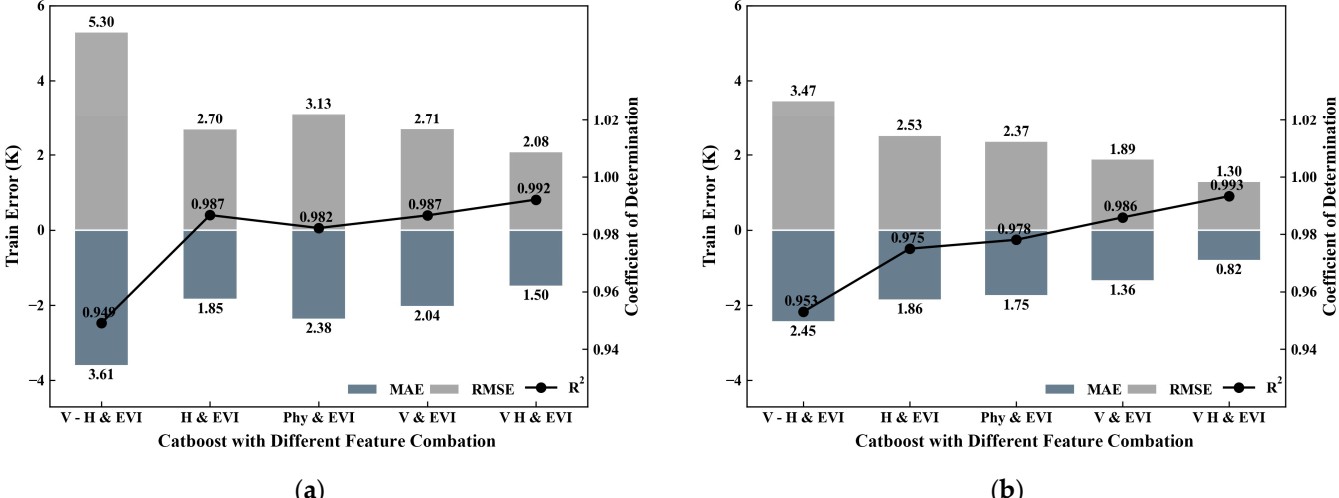

**Figure 5.** Results of 10–fold cross–validation of five combinations of feature vectors. (**a**) Daytime. (**b**) Nighttime.

Catboost uses a greedy machine learning approach to build decision trees by combining all features and to create nodes using all feature combinations. When the input feature vector of a model contains all the input feature vectors of another model, the Catboost sub–decision tree containing the most feature vectors considers the most feature combinations. This may be the reason why Catboost based on VH showed higher performance than the other models. In addition, the daytime performance of the Catboost model based on H (fully horizontal polarization) was higher than that of the other models. This further corroborates that the relatively discrete horizontal polarization of the correlation distribution may contribute the most to the spatial heterogeneity of LST.

### 3.3. Intercomparison and Analysis of LST Downscaling Results Based on Catboost

The AMSR–2 passive microwave BT data were resampled to a resolution of 1 km using the nearest neighbor method. These data, combined with MYD13A2 EVI data at 1 km resolution, were used as inputs for five models to retrieve the 1 km LST of Gurbantunggut Desert for the year 2019. The statistical approach of the LST downscaling model was based on the similarity of spatial structures at different scales. To validate the downscaled LST data, clear–sky pixels with a MYD11A1 quality average LST error of ≤1 K were selected. This allowed for further evaluation of the consistency and uncertainty of the downscaled LST data in terms of the disparities between them and MYD11A1 data across the five downscaling models.

In actual experiments, it is difficult to obtain the true surface temperature data at the pixel scale (1 km × 1 km downscaling) [2,16]. Wang et al. [63] obtained the true LST data for the period of 2000–2001 and evaluated the absolute retrieval accuracy of MODIS data. The results showed that the accuracy of the LST products was within 1 K. Yu et al. [64,65] validated the LST accuracy of MODIS data for Heihe River Basin and showed that the average absolute error was less than 2.2 K. Duan et al. [66] validated the accuracy of MODIS products for different underlying surfaces. Their results showed that the night RMSE for barren/sparsely vegetated areas was lower than the daytime one and that the average error was less than 2.75 K. The MODIS LST product can be used to indirectly assess the quality of other retrieval products. Therefore, the daytime and nighttime retrieval products of the Catboost models were compared with MODIS products. The evaluation indexes of the five downscaled model are reported below (Figure 6).

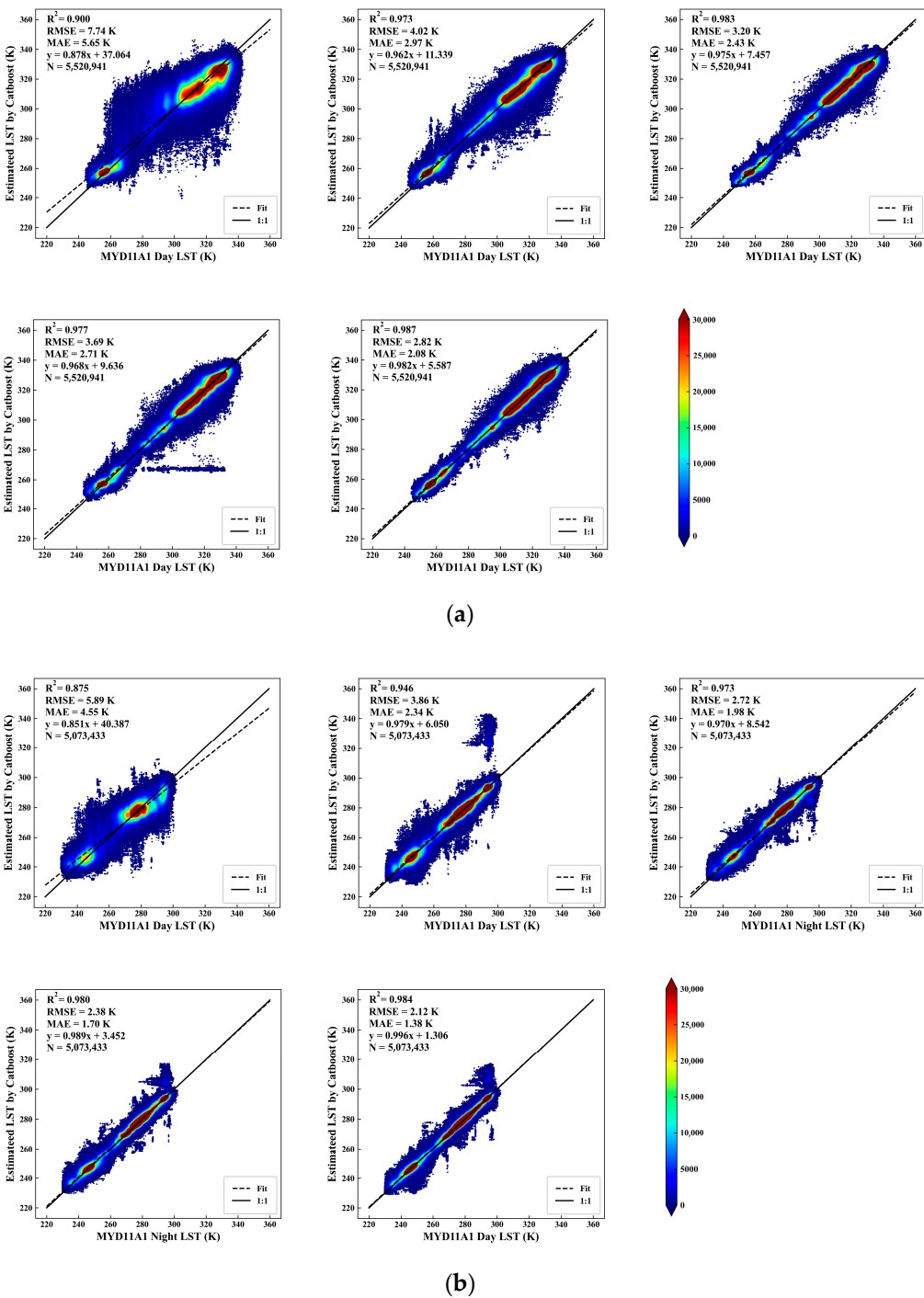

**Figure 6.** (**a**) Scatter diagram of downscaled LST obtained with daytime Catboost models (V–H/H/V/Phy/VH and EVI) against MYD11A data. (**b**) Scatter diagram of downscaled LST obtained with nighttime Catboost models (V–H/H/V/Phy/VH and EVI) against MYD11A data.

The a and b panels in Figure 6 show the scatter diagrams of downscaled LST obtained with Catboost against MYD11A1 data. The accuracy of daytime and nighttime models can be ranked, from the highest to the lowest accuracy, as VH and EVI > V and EVI > Phy and EVI > H and EVI > V–H and EVI, VH and EVI > Phy and EVI > V and EVI > H and EVI > V–H and EVI. Validation demonstrated that the Catboost–based VH and EVI model had the best level of agreement with MYD11A1, with daytime and nighttime $R^2$ of 0.987 and 0.984, RMSE of 2.82 K and 2.12 K, and MAE of 2.08 K and 1.38 K, respectively.

The accuracy rankings of the Phy and EVI model, and the V and EVI model, differed only in their performance during daytime and nighttime. This could be attributed to the high physical significance of the Phy and EVI feature vectors, which are able to mitigate the effects of atmospheric water vapor and surface water using channel differences. In contrast, the V and EVI combination was less physically significant but had stronger statistical significance. This was evident in its selection of vertically polarized vector factors that showed higher correlations with LST compared to other combinations. In the semi–heterogeneous underlying surface of Gurbantunggut Desert, temperature rises rapidly during the daytime, reaching its peak at around 13:00. The atmospheric water content decreases with increasing temperature, thereby weakening the advantage of the Catboost model based on Phy and EVI, which relies on strong physical significance, during the daytime. Conversely, during the nighttime, the temperature of the desert drops to its lowest level at around 1:00, causing an increase in atmospheric water content. At this time, the correction effect of the Catboost model based on Phy and EVI on water vapor can improve the accuracy of LST retrieval.

In our study, we conducted an analysis of the importance of each modeling factor, which is shown in Figure 7. This analysis is crucial for assessing the retrieval mechanism and gaining further insights into the statistical model. Daytime feature vector importance (importance > 5%): 7.3 GHz V(16.4) > 6.9 GHz V(15.4) > 89 GHz V(12.6) > EVI(12.4) > 89 GHz H(8.0) > 23.8 GHz V(6.1). Nighttime feature vector importance (importance > 5%): 89 GHz V(21.4) > 6.9 GHz V(13.4) > EVI(11.3) > 7.3 GHz V(10.2) > 89 GHz H(8.4) > 36.5 GHz V(6.3). Our analysis also revealed that the contribution of horizontally polarized BT to the model was low at all frequencies, except for the high frequency of the 89 GHz horizontally polarized channel.

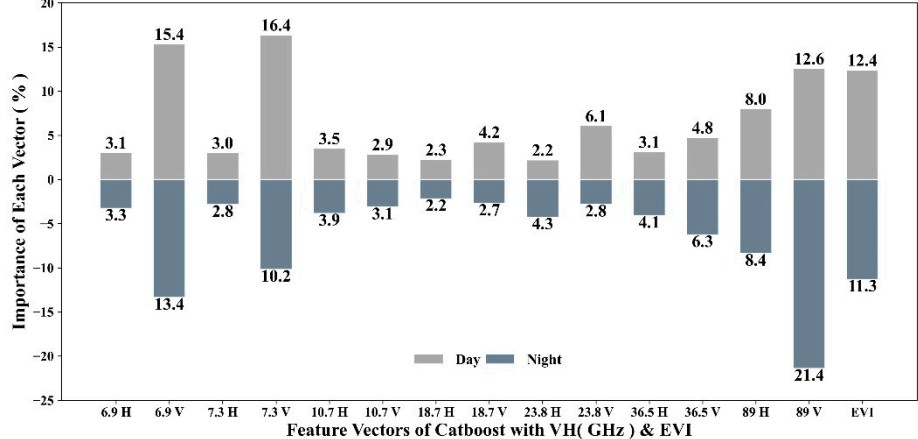

**Figure 7.** Importance bar graph of model (Catboost based on VH channels and EVI) factors for daytime and nighttime data.

The correlation statistics clearly differed from the importance results. For example, the daytime and nighttime correlation statistics of 23.8 GHz vertical polarization were the highest, while their importance accounted for a smaller percentage in the Catboost model. Daytime vertical polarization at 89 GHz had the lowest correlation, but it greatly contributed to the model according to the importance analysis (it ranked third). Nighttime vertical polarization at 6.9 GHz and 7.3 GHz had the lowest correlation statistics with respect to LST, but its importance rates in the Catboost model were all greater than 10%. The importance analysis indicted that both BT data at low (6.9 GHz and 7.3 GHz) and high (89 GHz) vertical polarization frequencies can provide more information to the Catboost model for the retrieval of LST. Correlation analysis may not be enough as the basis for Catboost model feature factor selection. Therefore, models based on all classical channel combinations were investigated in our article.

To summarize, our study demonstrated that the passive microwave surface temperature downscaling results obtained from the Catboost model were highly accurate, with the VH and EVI model exhibiting the highest accuracy. The accuracy of the retrieved LST during both daytime and nighttime was sufficient for regional surface temperature studies. The downscaled LST data we obtained compensated for the spatial discontinuity of MYD11A1 data. The daytime and nighttime LST products we generated can be used to analyze the spatial and temporal patterns of land surface temperature in the Gurbantunggut Desert.

### 3.4. Microwave Surface Temperature Correlation Analysis Based on Six–Layer Ground Temperature Data

In many current studies, there is a lack of quantitative analysis of depth about LST data retrieved using multi–channel BT due to the large differences in the penetration of different microwave channels. In this study, the correlation results were evaluated with r, p, and RMSE by conducting a correlation analysis on site multi–layer ground temperature data and the downscaled results (Figure 8). The daytime and nighttime downscaling results were highly significantly correlated (*p*-values $< 1.2 \times 10^{-100}$) with all six–layer ground temperature data. The highest correlation was found between the soil temperature of the 5 cm layer and the LST obtained using passive microwave downscaling.

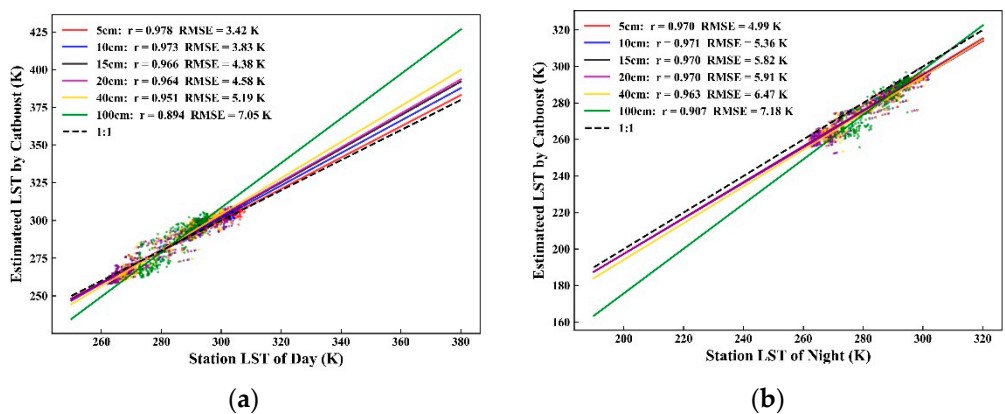

**Figure 8.** Correlation plots of Catboost VH ∣ EVI–downscaled LST against 6–layer station data. (**a**) Daytime. (**b**) Nighttime.

The results showed that both daytime and nighttime soil temperatures at all depths had a high correlation with the AMSR–2 downscaling results. RMSE tended to increase with the increase in the depth of the soil layer. Differences in the correlation between daytime– and nighttime–retrieved results mainly existed in four layers (5, 10, 15, and 20 cm). During the daytime, at the satellite transit time (about 15:00 BST), the soil temperature is significantly different in each layer due to surface net thermal radiation. It takes time for heat to be transferred to lower soil layers. Consequently, the LST of the surface layer was high, and that of the lower layers was low. At nighttime, at the satellite transit time (about 4:00 BST), the solar radiation energy acquired during the day by the upper surface has been transferred. We found that the soil temperature change with depth was small and that the variability was low. For the semi–heterogeneous underlying surface of Gurbantunggut Desert, the correlation coefficients of all soil layers were greater than 0.8, showing very strong correlation. This conclusion can be used to provide a basis for the passive microwave BT retrieval of the LST of multi–layer soil. LST retrieval based on passive microwaves can minimize the flaws of thermal infrared remote sensing due to strong cloud interference and obtain all–weather, daily, diurnal observations of six–layer soil temperature products.

### 3.5. Catboost–Based, Diurnal, All–Weather Surface Temperature Products

The results of our study demonstrate that the downscaled LST data obtained from the Catboost model, which was based on VH channels and EVI, are highly reliable. This

was confirmed by validation using site data, MYD11A1 data, and 10–fold cross–validation. The downscaled LST data were suitable for analyzing the spatial and temporal patterns of land surface temperature in the Gurbantunggut Desert. Our main goal was to develop a daily scale, 1 km spatially resolved LST product to compensate for the inability to use MYD11A1 data to obtain LST with cloudy pixels. Our study provides spatially continuous, diurnal–scale LST data. To fill the gaps in MYD11A1 data caused by clouds, we used LST data downscaled with AMSR–2 BT data. When the sky was clear, we used the LST pixel values provided by MODIS. When clouds or rain were present, we used AMSR–2 downscaled LST data. In cases where pixel values were still missing due to satellite scanning gaps, we utilized the GWR algorithm to fill them. Two consecutive days, 13 August and 14 August, are shown as examples in Figure 9.

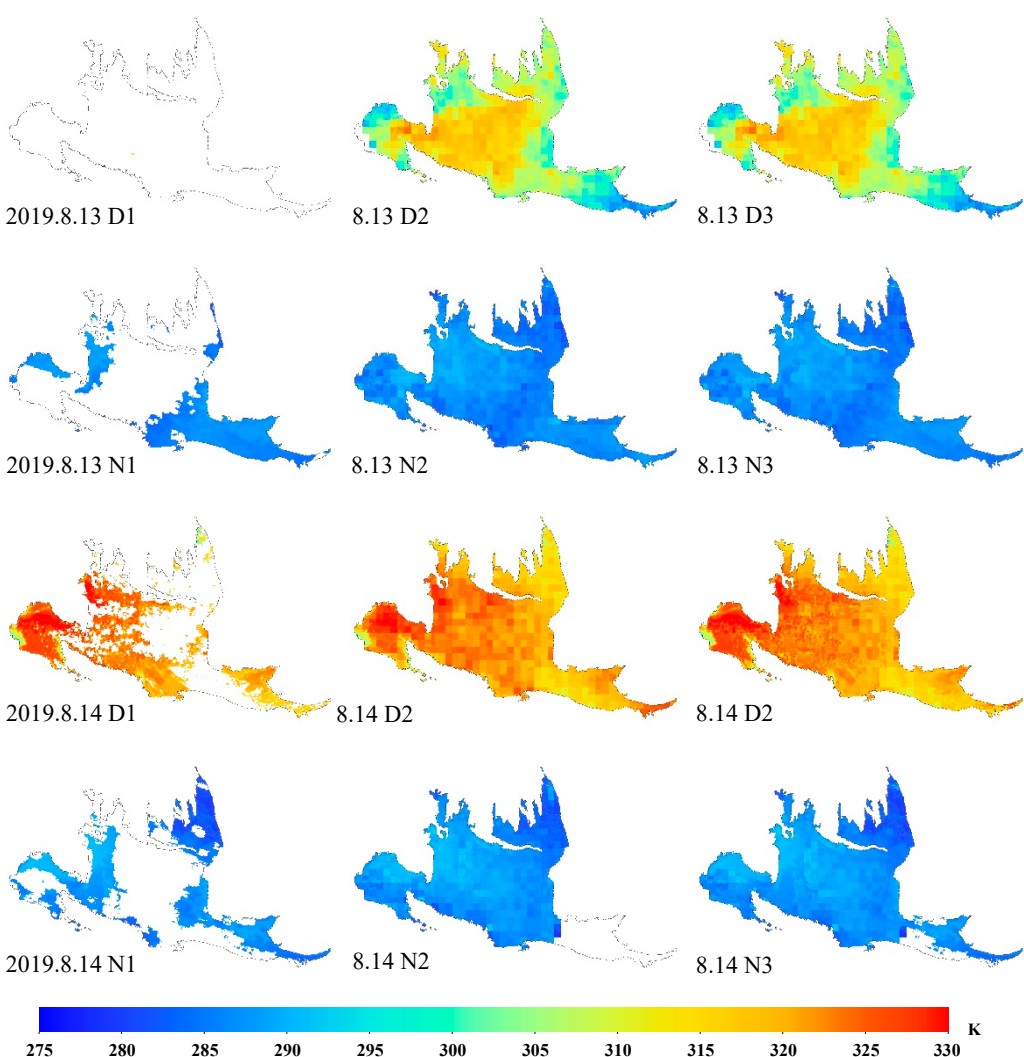

**Figure 9.** D represents day LST, and N represents night LST. (1) Cloud–free MODIS LST (K) at 1 km resolution, (2) AMSR–2–derived LST (K) at 1 km resolution, and (3) merged MODIS and AMSR–2 LST (K) at 1 km resolution.

## 4. Discussion

It is crucial to obtain all–weather, fine–scale surface temperatures using passive microwave downscaling models, and we assessed their applicability using Gurbantunggut Desert, which includes homogeneous and heterogeneous surface conditions composed of different rippled sand mounds and vegetation types. In this study, we evaluated the accuracy and mapping effectiveness of 1 km spatially resolved LST downscaled using passive microwave BT data of this semi–heterogeneous surface. Our results show that LST

downscaled using Catboost based on VH channels and EVI fitted well with MYD11A1 data, and its correlation with multi–layer soil temperature was analyzed. Overall, these results demonstrate the great potential of using the BT of passive microwaves to obtain the all–weather, fine–scale LST of a semi–heterogeneous underlying surface. Interestingly, some phenomena did appear that were not found in the downscaling work carried out on the mixed underlying surface.

The fusion of passive microwaves to obtain fine–scale and spatio–temporally continuous LST is feasible and holds potential [41,67]. Numerous studies have demonstrated the high accuracy of MODIS LST on different underlying surfaces, with precision ranging from 1 to 2.75 K [63–66]. Therefore, this study also conducted cross–validation of downscaled LST data using MYD11A1 products with an average LST error $\leq$ 1 K per pixel. Practical data from underlying surface stations can provide explicit and more intuitive indications of the accuracy of downscaled results. Therefore, our study selected data from the Fukang Desert Station, the only existing homogeneous underlying surface station in the Gurbantunggut Desert, to evaluate the accuracy of downscaled results. The spatial integrity of LST downscaled using the BT of passive microwave products is better than that of TIR LST [68,69]. Compared with the daytime RMSE of downscaled LST, the nighttime RMSE was low, showing a decrease of 0.6 K/1.57 K according to the validation of MYD11A1 or 5 cm soil temperature data, respectively. In addition, the accuracy of the downscaling results in this study area, i.e., RMSE values of 2.82 K (daytime) and 2.12 K (nighttime) according to the cross–validation of MYD11A1 data, is significantly higher than that calculated for other mixed underlying surfaces [41,70]. The reason for this phenomenon may be that the underlying surface in this study is relatively monotonous, and the machine learning algorithm of Catboost can adequately learn the relationship between the BT of passive microwaves and LST (the third paragraph of our discussion). In future work, we will expand the study area to carry out downscaled work on more complex mixed underlying surfaces and flow sand dunes with only terrain undulations. This will provide more practical data to verify the results. At the same time, we will explore the similarities and differences in the importance (contributions) of various vector factors during the downscaled process in multi–faceted underlying surfaces.

Correlation coefficient characteristics of factors of microwave BT and LST: Within the rectangular study area range (including the semi–homogeneous underlying surface of Gurbantunggut Desert and its surrounding oases), the statistical results of correlation (Figure 3) showed that the correlation coefficients between BT at 89 GHz and LST were higher than those of BT at lower frequencies, such as 23.8 GHz and 36.5 GHz. This result is consistent with the study by Mao Kebiao [32,33], whose findings represent the mainstream views [24,42]. However, regarding the semi–heterogeneous underlying surface of Gurbantunggut Desert, the correlation coefficients of BT at relatively lower frequencies were higher than those of BT at 89 GHz (Figure 3). It may be that vegetation, surface roughness, or other factors impact the correlation between BT and LST. This phenomenon further indicates that it is important for us to assess LST downscaling methods using passive microwaves on particular underlying surfaces of different complexities.

In our study, which involved the reconstruction of LST with cloudy pixels, we assumed that the relationship between LST and the BT of passive microwaves under clear–sky conditions could be applied to cloudy situations; then, the LST pixels of cloudy sky conditions were retrieved with the relationship. This reconstruction method with cloudy pixel information has also been widely used in other studies [71–74]. However, little research has been conducted on the applicability of this relationship to different underlying surfaces. We need to build a process–oriented evaluation method [75] to study the scale effects [76] of the relationship between downscaling factors and LST considering different underlying surfaces. This is what our next research will address.

## 5. Conclusions

In this study, we were able to successfully obtain downscaled LST data by utilizing the Catboost algorithm with multi–channel passive microwaves. Through our downscaling experiment, we have come to three main conclusions:

(1) The correlation coefficients between the feature vectors and LST of the semi–homogeneous underlying surface differed significantly from those of the surrounding oases, with the difference being more pronounced for daytime data. Specifically, the correlation coefficient of the semi–homogeneous underlying surface was high, while that of the surrounding oases was low. Moreover, we observed that the correlations between vertically polarized BT and LST were higher than those of horizontal polarization at the same frequency. As the frequency increased, the differences between the BT–LST correlation with horizontal polarization and that with vertical polarization at the same frequency became smaller.

(2) Our ten–fold cross–validation results revealed that the Catboost model based on VH exhibited the best stability, with daytime R2, MAE, and RMSE mean values of 0.992, 1.50 K, and 2.08 K, respectively, and nighttime R2, MAE, and RMSE mean values of 0.993, 0.82 K, and 1.30 K, respectively. These results indicate that the Catboost model based on VH established the mapping relationship between passive microwave BT and LST more accurately than the other four classical models.

(3) Validation using MYD11A1 data revealed that LST downscaled with the Catboost model based on VH and EVI had the highest accuracy, with daytime and nighttime R2 of 0.987 and 0.984, RMSE of 2.82 K and 2.12 K, and MAE of 2.08 K and 1.38 K, respectively. Furthermore, we validated the downscaled LST data using the six–layer soil temperature data of the site, which showed a highly significant, positive correlation with all six–layer soil temperature data of the site. However, the correlation coefficients (r) generally showed a decreasing trend with increasing depth, while RMSE showed an increasing trend.

**Author Contributions:** Conceptualization, Y.L. (Yongkang Li) and W.H.; methodology, Y.L. (Yongkang Li); software, Y.L. (Yongkang Li); validation, Y.L. (Yongkang Li); formal analysis, Y.L. (Yongkang Li), Y.L. (Yongqiang Liu), and Y.Y.; investigation, Y.L. (Yongqiang Liu); resources, Q.H.; data curation, Y.Y.; writing—original draft preparation, Y.L. (Yongkang Li); writing—review and editing, W.H., Q.H. and Y.Y.; visualization, Y.L. (Yongkang Li); supervision, Y.L. (Yongqiang Liu) and J.T.; project administration, Q.H.; funding acquisition, Q.H. All authors have read and agreed to the published version of the manuscript.

**Funding:** This work was supported by the Second Tibetan Plateau Scientific Expedition and Research (STEP) Program (grant no. 2019QZKK010206–02) and XJU2022BS056.

**Data Availability Statement:** The data that support the findings of this study are available from the corresponding author upon reasonable request.

**Acknowledgments:** We sincerely thank Fukang Desert Ecology Experiment Station, Xinjiang Institute of Ecology and Geography, and the Chinese Academy of Sciences for providing data, and Lei Cheng for his help. We thank Jing Zhao of the Institute of Air and Space Information Innovation, Chinese Academy of Sciences, for her help in revising the paper.

**Conflicts of Interest:** The authors declare no conflict of interest.

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
